# Clinical Features and Prognostic Impact of Pancreatic Ductal Adenocarcinoma without Dilatation of the Main Pancreatic Duct: A Single-Center Retrospective Analysis

**DOI:** 10.3390/diagnostics13050963

**Published:** 2023-03-03

**Authors:** Takuya Takayanagi, Yusuke Sekino, Noriki Kasuga, Ken Ishii, Hajime Nagase, Atsushi Nakajima

**Affiliations:** 1Department of Gastroenterology, Yokohama Rosai Hospital, Yokohama 222-0036, Japan; 2Department of Gastroenterology and Hepatology, Yokohama City University Hospital, Yokohama 236-0004, Japan

**Keywords:** main pancreatic duct, pancreatic ductal adenocarcinoma, endoscopic ultrasound

## Abstract

The presence of main pancreatic duct (MPD) dilatation is important for diagnosing pancreatic ductal adenocarcinomas (PDACs). However, we occasionally encounter PDAC cases without MPD dilatation. The objectives of this study were to compare the clinical findings and prognosis of pathologically diagnosed PDAC cases with and without MPD dilatation and to extract factors related to the prognosis of PDAC. The 281 patients pathologically diagnosed with PDAC were divided into two groups: the dilatation group (*n* = 215), consisting of patients with MPD dilatation of 3 mm or more, and the non-dilatation group (*n* = 66), consisting of patients with MPD dilatation less than 3 mm. We found that the non-dilatation group had more cancers in the pancreatic tail, more advanced disease stage, lower resectability, and worse prognoses than the dilatation group. Clinical stage and history of surgery or chemotherapy were identified as significant prognostic factors for PDAC, while tumor location was not. Endoscopic ultrasonography (EUS), diffusion-weighted magnetic resonance imaging (DW-MRI), and contrast-enhanced computed tomography had a high tumor detection rate for PDAC even in the non-dilatation group. Construction of a diagnostic system centered on EUS and DW-MRI is necessary for the early diagnosis of PDAC without MPD dilatation, which can improve its prognosis.

## 1. Introduction

Pancreatic ductal adenocarcinoma (PDAC) is the most common type of pancreatic cancer (PC), and the number of patients with PDAC is increasing [1,2]. According to the Vital Statistics from the Ministry of Health, Labour, and Welfare, 36,356 patients died of PC, and it was deemed the fourth leading cause of cancer-related deaths in Japan in 2019 [3]. PDAC is among the most lethal malignancies, with 5-year survival rates as low as 10% and 8.5% in the United States and Japan, respectively [1,2,4]. The American Cancer Society estimated that in 2021, 60,430 patients would be diagnosed with PC, leading to the death of 48,220 patients [5]. The number of deaths due to PC is expected to rise to 63,000 in the United States by 2030, making it the second leading cause of cancer-related deaths [6]. 

The presence of main pancreatic duct (MPD) dilatation is important in diagnosing PC [7]. Most PDACs are thought to originate from the branch of pancreatic ducts near the MPD [8], and even small PDACs have a high rate of extension and invasion into the MPD. A study reported MPD dilatation in approximately 80% of 200 early PDAC cases [9]. However, we occasionally encounter PDAC cases without MPD dilatation, even in advanced cancers such as those located in the groove, pancreatic uncus, or the most caudal side of the pancreas. Therefore, a detailed comparison between PDAC cases without MPD dilatation and those with MPD dilatation would provide valuable information for the development of strategies for the early diagnosis of PDAC. However, our literature review revealed few reports on the clinical findings of PDAC without MPD dilatation. Therefore, we analyzed pathologically diagnosed PDAC cases with and without MPD dilatation and compared their clinical findings and prognoses. In addition, we extracted factors related to the prognosis of PDAC.

## 2. Materials and Methods

### 2.1. Patients

This is a retrospective, single-center observational study. Patients with PC were registered in a hospital-based cancer registry, which is part of the National Cancer Registry [10] at Yokohama Rosai Hospital, between April 2014 and December 2021. Patients with suspected cancer were identified and aggregated into a single list based on disease type, pathological findings, therapy (surgical, chemotherapy, palliative care, radiation) history, and history of referral to cancer centers. The patients’ records were then searched to collect detailed information. PC was defined according to the International Statistical Classification of Diseases and Related Health Problems, Tenth Revision (ICD–10) code C25. Among the patients, only those with pathologically diagnosed PDAC in our hospital were included in this study. However, patients with postoperative recurrence and those with intraductal papillary mucinous neoplasm (IPMN)-derived invasive cancer showing a histologic transition between IPMN and PDAC were excluded. Patients were considered eligible if they were diagnosed or started their first treatment (including palliative care) at the study hospital. They were considered ineligible if they were diagnosed and started their first treatment at other hospitals. The date of diagnosis was defined as the date of the pathological diagnosis. MPD dilatation was defined as a maximum MPD diameter of ≥3 mm on ultrasonography (US), computed tomography (CT), magnetic resonance imaging (MRI) (magnetic resonance cholangiopancreatography [MRCP]), or endoscopic US (EUS) imaging modalities, regardless of the site [11]. PDAC was defined according to the clinical stage based on the National Comprehensive Cancer Network Clinical Practice Guidelines in Oncology (version 1, 2020) [12] and resectability classification based on the 8th edition of the Union for International Cancer Control [13]. 

### 2.2. Evaluations

We divided the eligible PDAC cases into two groups: the dilatation group, consisting of patients with MPD diameter of 3 mm or more, and the non-dilatation group, consisting of patients with MPD diameter of less than 3 mm, and compared the following.: (1) patient characteristics, (2) reasons for medical examination, (3) blood tests and imaging findings, (4) pathological examination, and (5) prognosis. In addition, we extracted factors related to the prognosis of PDAC.

### 2.3. Imaging and Pathological Diagnosis

#### 2.3.1. Imaging Diagnosis

Depending on the facilities available at each institution, US was performed by ultrasonographers certified by the Japan Society of Ultrasonics in Medicine using Aplio i700 (Canon Inc. Tokyo, Japan) or LOGIQ E10 (GE Healthcare, Tokyo, Japan).

CT was performed using a multidetector row from 64 to 320 slices (Aquilion ONE, Canon Inc. Tokyo, Japan). A contrast medium of 300 mg I/mL is used, and the contrast agent (Iopamidol, Bayer Yakuhin, Ltd., Osaka, Japan or Iohexol, GE Healthcare, Tokyo, Japan) was injected in 30 s at a dose of 2 mL (maximum volume 150 mL) per body weight (kg). Four phases (plain, arterial, portal, and equilibrium) were imaged, and three-dimensional CT angiography images were created from the 1 mm horizontal section images of these phases, the anterior forehead section images of the portal phase, and the data from the arterial phase.

MRI (MRCP) was performed using either a 1.5 Tesla magnet (EXCELART Vantage Powered by Atlas, Canon Inc. Tokyo, Japan) or a 3.0 Tesla magnet (MAGNETOM Skyra, Siemens Healthineers AG., Erlangen, Germany) with a surface phased-array coil. The examinations were performed on fasted patients in the supine position who drank 1200 mg of FerriSeltz powder (Otsuka Pharmaceutical Co., Ltd., Tokyo, Japan). T2-weighted and non-contrast-enhanced images were obtained and used for three-dimensional reconstruction.

The source images from the thin collimation multislice acquisition, three-dimensional reconstruction, and axial, coronal, and sagittal oblique planes were interpreted by two designated doctors who were board-certified gastroenterologists of The Japanese Society of Gastroenterology with more than 10 years of experience in clinical practice.

For EUS, all patients were administered midazolam intravenously before the procedure, and their heart rate, blood pressure, and peripheral oxygen saturation were monitored during the procedure. The EUS procedures were performed by one of three experienced endosonographers who had performed >300 EUSs each. EUS was performed using a curved linear-array echoendoscope (GF–UCT260, Olympus Medical Systems Corp., Tokyo, Japan) paired with an ultrasound system (EU–ME2 premium, Olympus Medical Systems Corp.).

In addition, CT and MRI (MRCP) findings were reviewed by two Japanese Society of Gastroenterology specialists who had been in practice for at least 10 years, while EUS imaging was reviewed by one board-certified member of the Japanese Society of Pancreatology.

#### 2.3.2. Pathological Diagnosis

For the pathological examination, we performed endoscopic retrograde cholangiopancreatography (ERCP) using a video duodenoscope (JF–260V, Olympus Medical Systems Corp., Tokyo, Japan). When obstructive jaundice due to distal bile duct stricture caused by a pancreatic tumor was suspected, cholangiography was performed after bile duct intubation, and a guidewire (VisiGlide2 0.025 inch, Olympus Medical Systems; Jagwire^TM^ 0.035 inch; Boston Scientific, Marlborough, MA, USA) was inserted. Brush cytology (RX Cytology Brush, Boston Scientific) and forceps biopsy (Radial Jaw 4P, Boston Scientific) were performed on the distal bile duct stricture, and a 6-Fr endoscopic nasobiliary drainage (ENBD Tube for nasal drainage, Gadelius Medical, Tokyo, Japan; NB–Braid pig tail, PIOLAX Medical Device, Yokohama, Japan) catheter was placed for biliary drainage and biliary juice cytology. In contrast, if a pancreatic tumor was suspected of causing MPD stenosis or dilatation, pancreatography was performed, a guidewire was inserted, and brush cytology was performed in the area of the MPD stenosis. An endoscopic nasopancreatic drainage (ENPD) catheter was implanted for serial pancreatic juice aspiration cytologic examination (SPACE). We used a 5-Fr ENPD catheter (Nasal Pancreatic Drainage Set, Cook Medical, Tokyo, Japan) to collect pancreatic juice twice a day for up to six times and subsequently removed the ENPD catheter. The brush was inserted into the common bile duct or MPD of interest over the guidewire and was positioned distal to the stricture. It was advanced from the sheath to a point proximal to the stricture and moved across the stricture in a to-and-fro manner 15–20 times. Subsequently, biliary or pancreatic juice in the catheter was flushed with saline for collection.

When obvious pancreatic lesions were observed on EUS, EUS–FNA was performed using a linear echoendoscope (GF–UCT260, Olympus Medical Systems Corp., Tokyo, Japan) with a 22- or 25-gauge needle (Acquire^TM^, Boston Scientific; EZ Shot 3 Plus, Olympus Medical System; EchoTip^TM^, Cook Medical). 

These examinations were performed under the supervision of specialists with experience in performing more than 100 EUS–FNA and 500 ERCP procedures for more than 10 years.

When pancreatic tumor invasion was suspected in the gastrointestinal tract, such as the stomach or duodenum, a biopsy was performed using forceps (Radial Jaw 4P, Boston Scientific). Percutaneous needle biopsy (Quick–Core 19 G, Cook Medical) was performed under echo guidance when a metastatic liver tumor was suspected. Percutaneous needle puncture cytology was performed under echo guidance when ascites were present.

### 2.4. Statistical Analysis

Statistical analysis was performed using the JMP v14.0 software (SAS Institute, Chicago, IL, USA). For two-tailed tests, Pearson’s χ^2^ test was used to identify statistically significant differences. *p*-values < 0.05 were considered statistically significant. We used Kaplan–Meier survival analysis to compare the overall survival (OS) between the two groups. Differences in survival were evaluated using the log-rank test. We performed a multivariate analysis using the Cox proportional hazards model for factors significantly associated with OS in the univariate analysis.

## 3. Results

### 3.1. Patient Characteristics

Figure 1 shows the patient flow diagram. During the study period, 355 patients with PC were registered in the hospital-based cancer registry at Yokohama Rosal Hospital, 296 of whom had pathological evidence of PC. Pathological evidence was defined as preoperative diagnosis by cytology and histology using ERCP, EUS-FNA, etc., and postoperative pathology in surgical specimens. We excluded 59 patients who were diagnosed with PC based on imaging findings and tumor markers without pathological confirmation. Of the 296 patients with pathological evidence of PC, 7 with intraductal papillary mucinous carcinoma, 5 with metastatic PC, 2 with neuroendocrine carcinoma, and 1 with postoperative recurrence were excluded. The remaining 281 patients pathologically diagnosed PDAC were divided into two groups: the dilatation group (*n* = 215, 76.5%) for cases with MPD dilation to 3 mm or more and the non-dilatation group (*n* = 66, 23.5%) for cases with MPD dilation less than 3 mm (Figure 1).

Table 1 shows a comparison of the clinical characteristics between the dilatation and non-dilatation groups.

There was a significant difference in tumor location between the two groups, with the dilatation group having more tumors in the pancreatic head and the non-dilatation group having more tumors in the pancreatic tail. There were no significant differences in the sex ratio, age, tumor size, and risk factors between the groups.

### 3.2. Reasons for Medical Examination

Table 2 shows a comparison of the opportunities for the medical examination between the dilatation and non-dilatation groups. 

In the dilatation group, jaundice was the diagnostic opportunity in significantly more cases, whereas weight loss was the diagnostic opportunity in significantly more cases in the non-dilatation group.

There were no significant differences in abnormalities identified on medical check-up, during examination or follow-up for other diseases, or during follow-up for pancreatic diseases between the two groups.

### 3.3. Imaging Findings

Table 3 shows a comparison of the diagnostic imaging data between the dilatation and non-dilatation groups. The direct detection rate of tumors using US was significantly lower in the non-dilatation group than in the dilatation group. Furthermore, EUS, diffusion-weighted magnetic resonance imaging (DW-MRI), and contrast-enhanced CT (CECT) had a high direct tumor detection rate for PDAC with or without MPD dilatation among the modalities. Regarding indirect imaging findings, MPD dilatation was observed in 70% (106/151) of patients using US, 95% (197/207) using CECT, 92% (122/133) using MRI (MRCP), and 90% (161/179) using EUS. Meanwhile, MPD stenosis was detected more frequently in the dilatation group than in the non-dilatation group in all modalities (Table 3).

### 3.4. Pathological Diagnosis

Table 4 shows a comparison of the histopathological diagnostic data between the dilatation and non-dilatation groups. We performed ERCP on 145 patients (52%) and EUS–FNA on 199 patients (71%). The diagnostic sensitivities of ERCP and EUS–FNA were 62% and 93%, respectively. There was a significant difference in the number of ERCP procedures performed between the dilatation and non-dilatation groups. In contrast, there was no significant difference in the diagnostic sensitivities of gastrointestinal biopsy, liver tumor biopsy, or ascites cytology between the two groups (Table 4).

### 3.5. Clinical Stage and Resectability Classification

Table 5 show a comparison of the dilatation and non-dilatation groups in terms of clinical stage and resectability classification, respectively. 

Clinical stage IIA was significantly more common in the dilatation group, while clinical stage IV was significantly more common in the non-dilatation group (Table 5A). As for the resectability classification, borderline resectable with portal vein invasion (BR-PV) was significantly more common in the dilatation group, while unresectable with metastasis (UR-M) was significantly more common in the non-dilatation group (Table 5B).

#### 3.5.1. Prognosis

Figure 2 shows the Kaplan–Meier estimates of the OS of PDAC cases in the dilatation and non-dilatation groups. The OS was significantly longer in the dilatation group than in the non-dilatation group; the median survival time was 230 days in the dilatation group and 88 days in the non-dilatation group (*p* = 0.001). 

#### 3.5.2. Factors Affecting PDAC Prognosis

We also evaluated the factors associated with OS in 281 pathologically diagnosed PDAC cases. In the univariate analysis, tumor location, clinical stage, resectability classification, treatment, and the presence of MPD dilatation were significantly associated with OS (Table 6). Furthermore, clinical stage IV and best supportive care (BSC) were extracted as significant factors associated with OS in a multivariate analysis. These results indicate that the clinical stage at diagnosis and its corresponding treatment, rather than the MPD dilatation itself and tumor location, determine prognosis.

## 4. Discussion

The objectives of this study were to compare the clinical findings and prognosis of pathologically diagnosed PDAC cases with and without MPD dilatation and to extract factors related to the prognosis of PDAC.

The major findings of this study are as follows: First, PDAC without MPD dilatation was more commonly located in the pancreatic tail and had more advanced disease stage, lower resectability, and worse prognoses than PDAC with MPD dilatation. Second, among the symptoms that triggered opportunities for medical examination, jaundice was more common in patients with PDAC with MPD dilatation while weight loss was more common in patients with PDAC without MPD dilatation. Third, the tumor direct detection rate using US was significantly lower in PDAC cases without MPD dilatation than in those with MPD dilatation; additionally, EUS, DW-MRI, and CECT had a high direct tumor detection rate of PDAC in both groups. Fourth, Clinical stage IV and BSC were extracted as significant prognostic factors for PDAC, while the presence or absence of MPD dilatation and tumor location were not.

The possible reason why the pancreatic tail is the most common site of PDAC without MPD dilatation is this: unlike in the head and body, the length of the MPD in the pancreatic tail is short; hence, when a tumor develops in the pancreatic tail and invades the MPD, it fills the dilatated MPD as it grows, thus decreasing its diameter.

Meanwhile, the worse prognosis of the non-dilatation group may be explained by the clinical stage at the time of diagnosis, which was worse in the non-dilatation group than in the dilatation group; furthermore, many patients were unable to choose treatment such as surgery or chemotherapy and were managed with BSC (Table 6). Therefore, we believe that early diagnosis of PDAC without MPD dilatation is necessary to improve the prognosis of PDAC as a whole. It has been reported that approximately half of the cancers detected by weight loss were clinical stage IV [14]. This means that many of the cancers detected as a result of weight loss are advanced. The reason for the higher frequency of weight loss in the non-dilatation group may be that the percentage of clinical stage IV cancers is higher in the non-dilatation group than in the dilatation group, because it is also more difficult to find early stage cancer in the non-dilatation group. Additionally, 39% of the patients in the non-dilatation group had no symptoms. This indicates that PDAC without MPD dilatation may be detected incidentally. The asymptomatic onset of PDAC without MPD dilatation may lead to a delay in diagnosis, and as a result, patients may end up with an advanced stage (i.e., clinical stage IV) at diagnosis due to the gradual progression of the disease. In addition, this may explain the worse condition of patients with PDAC without the MPD dilatation compared to those with the MPD dilatation. Concerning the mechanism of MPD dilatation in PC, mechanical compression or invasion by the tumor may cause segmental obstruction and upstream dilatation in the MPD [15]. Therefore, it is considered that most PC originate from the branch of pancreatic ducts near the MPD then grow towards the MPD [15,16]. In a multicenter study on early-stage PDAC in Japan, MPD dilatation could be detected in approximately three-quarters of the Stage 0 cases (76.5% using US, 72.0% using CT, and 73.9% using MRI) [9]. However, in the present study, PDAC without MPD dilatation was present in 23.5% of all PDAC cases. In 2009, Kanno et al. [17] cited and examined the differences in development and pancreatic ductal extension at the PC site using hamsters reported by Tsutsumi [18]. According to the report, cancers arising from precancerous lesions in relatively large pancreatic ducts begin to invade the pancreatic ducts after their extension, and as the invasive area increases, the intraductal elements are destroyed. Furthermore, lesions arising in the peripheral pancreatic ducts have a high proliferative capacity from the initial stage of onset and tend to develop rapidly into invasive cancer without progressing through the pancreatic duct. Therefore, PC arising from the branch of pancreatic ducts near the MPD may show intraepithelial extension and cause changes in the pancreatic duct. In contrast, PC arising from the peripheral branches may be detected in an advanced state because they invade the surrounding area without intraepithelial extension. We believe that most cases of PDAC without MPD dilatation in this study correspond to the latter mechanism. 

We had expected that the PDAC of the pancreatic tail would be a significant prognostic factor since a large proportion of these patients do not exhibit MPD dilatation; moreover, the disease is often detected at an advanced stage due to the lack of a diagnostic trigger, such as jaundice. In this study, as shown in Table 1, 40 out of 55 cases of pancreatic tail adenocarcinoma were not accompanied by MPD dilatation. However, the multivariate analysis shown in Table 6 revealed that only Stage IV and BSC, not tumor location, were significant prognostic factors. This may be due to the increasing number of asymptomatic cases of PDAC diagnosed at an early stage with the availability of EUS, DW-MRI, and CECT; in these cases, where neither the tumor location nor the presence or absence of MPD dilatation was a prognostic factor. The diagnosis of PDAC with or without MPD dilatation at an earlier stage is important to improve prognosis. 

As shown in Table 3, it is difficult to diagnose PDAC without MPD dilatation by US because of the poor reliability of indirect findings; however, EUS, DW-MRI, and CECT can be expected to identify these lesions well, as is the case for PDAC with MPD dilatation. However, because of the radiation exposure problem with CT, EUS and DW-MRI play an important role in terms of detecting these lesions.

In this study, we examined the classification of cancers of the pancreatic head, body, and tail based on the General Rules for the Study of Pancreatic Cancer (7th Edition) [19]. Herein, we propose a new clinical classification of PDAC without MPD dilatation, as described below (Figure 3). 

Type I: No mass can be detected using any imaging modality. The MPD is narrowed, localized pancreatic atrophy is observed, or EUS shows a hypoechoic area surrounding the MPD stenosis. All locations are considered acceptable. A typical example is a carcinoma-in situ (Figure 4).

Type II: Cases that form a mass located in the pancreatic uncus region (Figure 5). 

Type III: Cases that form a mass located in the groove region (Figure 6). 

Type IV: Cases with a mass outside the groove region, pancreatic uncus, and the most caudal part of the pancreas (Figure 7).

Type V: Cases that form a mass located on the caudal side of the pancreas (Figure 8). 

Table 7 shows the classification of the 66 cases of PDAC without MPD dilatation in our study using the above classification method. Among the patients with PDAC without MPD dilatation, type V was the most common. Cases of bile duct obstruction were most common in type III followed by type II, and cases of duodenal obstruction were most common in type II followed by type III (Table 7).

Basing on the results of the present study and our new classification (Figure 3), we propose a diagnostic management algorithm of PDAC without MPD dilatation in Figure 9. After carrying out MRI/MRCP, EUS, and CT and PDAC without MPD dilatation is suspected, it is divided based on our proposed classification system. Endoscopic retrograde pancreatography (ERP) is recommended for type I, endoscopic retrograde cholangiography (ERC) and EUS-FNA for types II–IV if obstructive jaundice is present, EUS-FNA alone if no obstructive jaundice is present, and EUS-FNA for type V (Figure 9).

US is an important first-step imaging modality in pancreatic examinations [20]. It has been reported that a slight dilatation of the MPD and pancreatic cysts detected using US are important predictive signs. Tanaka et al. diagnosed 12 cases of PC and stages 0 and I of 1058 prospective follow-up cases with these predictive signs and recommended periodic checks in these cases [21]. In the present study, the direct tumor detection rate was 85%, the MPD dilatation detection rate was 70%, and the MPD stenosis rate was 59% in the dilatation group, which is sufficient for the first step of close examination. However, in the non-dilatation group, the direct tumor detection rate was 42%, and the MPD stenosis rate was 0%, which is an inadequate result. This may be because in most cases that do not show dilatation of the MPD, the tumor was located in the most caudal part of the pancreas, the groove region, or the pancreatic uncus, as mentioned above, all of which are difficult to delineate using US [22].

EUS is superior to CT and US in terms of spatial resolution, making it an essential modality for diagnosing pancreatic tumors [15,16]. In the present study, EUS enabled a significantly higher tumor detection rate in patients with PDAC with or without MPD dilatation than CT or MRI. Furthermore, several cases were observed in which tumors could not be detected using CT or MRI; however, they could be detected using EUS. It has been reported that EUS has a diagnostic sensitivity of 94.4% for detecting small PDAC (<20 mm) [23]. Yasuda et al. reported that of 132 patients with risk factors for PDAC without masses detected on CT, pancreatic tumors were detected in three patients using EUS [24]. Regarding PDAC without MPD dilatation, only a few cases showed indirect findings. It is known that cases in the pancreatic tail, which corresponds to the type V we proposed, lack evidence of pancreaticobiliary stenosis and are often advanced in stage at the time of detection [25]. Therefore, it is important to detect direct findings of PDAC, including type V cases, without indirect findings such as dilatation of the MPD. Therefore, we believe that with its high direct detection rate, EUS should be aggressively performed in patients with high-risk factors for PC who do not show indirect findings on US, CT, or MRI (MRCP). In performing EUS, it is very important to observe the uncus (type II), groove (type III), parenchymal margin (type IV), and most caudal side (type V) of the pancreas rather than just observing the surroundings of the MPD (type I).

Recently, localized pancreatic atrophy, among the secondary findings of PDAC, has been reported to be important for the early diagnosis of PDAC [26]. Nakahodo et al. reported that histopathologically localized pancreatic atrophy occurred around high-grade PanIN lesions in patients without invasive cancer; in 63% (17/27) of these cases [27]. Kobashi et al. named localized pancreatic atrophy “K-sign” and reported that 24 of 41 patients (58.5%) had “K-sign” on preoperative CT images [28]. Toshima et al. analyzed abnormalities of the pancreas on CT performed at least 1 year before the diagnosis of clinical stage I PDAC; they showed that a focal pancreatic abnormality was present on the most recent pre-diagnostic CT images in 55/103 (53.4%) patients with PDAC and that the most common focal abnormality was atrophy, which accounted for 39/103 (37.9%) cases [29]. Miura et al. reported that 9/41 (22.0%) patients presented with localized pancreatic atrophy on CT performed at 2 and 3 years before PDAC diagnosis, but none presented with MPD changes, suggesting that localized pancreatic atrophy presents earlier than changes in the MPD [30]. In fact, one case of type I in our study clearly showed localized pancreatic atrophy but little change in the MPD (Figure 4).

Pathological diagnosis is very important in the diagnosis of PC. It is also significant to obtain histological evidence before treatment due to the recent increase in the need for neoadjuvant chemotherapy for PDAC. EUS-FNA is now widely used for the pathological diagnosis of solid pancreatic lesions due to its high diagnostic performance and safety. A meta-analysis of EUS-FNA for pancreatic tumors showed pooled sensitivity of 86.5–90.2% and pooled specificity of 95.5–98% [31,32,33,34], demonstrating its effectiveness. In particular, the rate of positive diagnosis for tumor diameters of 10 to 20 mm is 83.5–95%, and when limited to lesions of 10 mm or less, 82.5–96.0% [35,36]. EUS-FNA also plays an important role in PC without MPD dilatation. In particular, EUS-FNA is necessary to obtain pathological evidence because ERCP cannot be used to obtain specimens in patients who do not have obstructive jaundice.

ERCP is an important tool in the diagnosis and treatment of PC. The utility of ERCP-associated pancreatic juice cytology (PJC), especially SPACE, has been reported using an ENPD catheter [37,38,39,40,41]. Ikemoto et al. reported that while a single PJC showed a sensitivity of 38%, SPACE showed a significant improvement, with its sensitivity reaching 75% and rising to 83% when considering only patients in stage 0 [42]. In the present study, the sensitivity of SPACE in the dilatation group was 56%, which is slightly lower than that previously reported; however, it was higher than the sensitivity of single aspiration of pancreatic juice, which was at 41%. In the non-dilatation group, the number of cases that showed MPD stenosis was low, and the number of SPACE procedures were limited. Therefore, ENPD should be used in cases of localized stenosis and distal dilatation of the MPD during ERCP [43].

Currently, liquid biopsy, e.g., circulating tumor DNA, has emerged as a promising prognostic biomarker of PDAC [44,45,46]. Circulating tumor DNA has gained popularity for cancer diagnostic, prognostic, or therapeutic monitoring applications since its identification in the serum of cancer patients [47]. Non-invasive early-stage pancreatic cancer develops with mutations in KRAS, and these PDAC precursor lesions are thought to progress to invasive cancer through the inactivation of tumor suppressor genes such as TP53, SMAD4, and CDKN2A4, following the KRAS mutation [48]. In order to establish more advanced techniques for the early diagnosis of PDAC, progress in liquid biopsy research will be essential to compensate for the limitations of imaging techniques [49]. 

It has been reported that regional networks between specialists In PC (SPC) and general practitioners (GP) should play an important role for the early diagnosis of PC. Onomichi city is a rural city located in Hiroshima Prefecture in western Japan, and its total population is approximately 150,000. Onomichi General Hospital and Onomichi Medical Association established a community program for the early diagnosis of PC in 2007. From January 2007 to June 2014, a total of 6475 cases consulted SPC after starting this program. As a result of this project, GP are able to pick up high-risk cases of PC and smoothly link them to a precise SPC examination, resulting to 399 out of 6475 cases being histologically diagnosed with PC. Of these cases, 16 were finally diagnosed as carcinoma in situ [50]. As the concept of the Onomichi project spreads, some Japanese medical associations have tried to establish the regional network for the early diagnosis of PC. As shown in Table 3, EUS has a very favorable diagnostic performance without radiation exposure even in the non-dilatation group. We hope that EUS practice, considering the site of onset of PC without MPD dilatation in Figure 3, will lead to early diagnosis of PC and improvement of patient prognosis. Therefore, EUS should be performed at an earlier stage in the diagnosis of PC, including cases without MPD dilatation, as an examination for high-risk cases of PC such as those with family history of PC, chronic pancreatitis, and IPMN.

This study has some limitations, including its retrospective, single-center, and observational nature. Our hospital is a high-volume center for pancreatic diseases and a city hospital. This major city hospital performs approximately 900 observational EUS cases, 120 EUS-FNA cases, 25 interventional EUS cases, and 700 ERCP cases annually. In addition, many patients included in this study were referred from other hospitals; therefore, the referring physicians might have had a selection bias. 

## 5. Conclusions

PDAC without MPD dilatation was more common in the pancreatic tail, had worse prognosis, a more advanced disease stage, and lower resectability than PDAC with MPD dilatation. Although PDAC without MPD dilatation had the most cases in the pancreatic tail, tumor location was not a significant prognostic factor for PDAC; rather, clinical stage and treatment (specifically, whether surgery or chemotherapy was performed) were extracted as significant factors. EUS, DW-MRI, and CECT had a high direct tumor detection rate for PDAC with or without MPD dilatation among all modalities. Construction of a diagnostic system centered on EUS and DW-MRI is necessary for the early diagnosis of PDAC without MPD dilatation, which can improve PDAC prognosis.

## Figures and Tables

**Figure 1 diagnostics-13-00963-f001:**
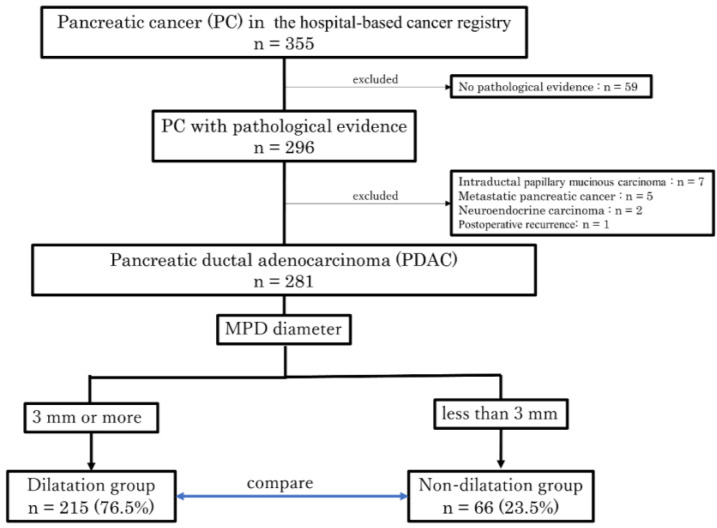
Patient selection flow diagram. A flow diagram illustrating the process of enrolling and selecting patients with PDAC for this study. A total of 281 patients were divided into the dilatation group, consisting of cases with MPD dilation to 3 mm or more, and the non-dilatation group, consisting of cases with MPD dilation less than 3 mm. After that, the two groups were compared for the set outcomes. MPD, main pancreatic duct.

**Figure 2 diagnostics-13-00963-f002:**
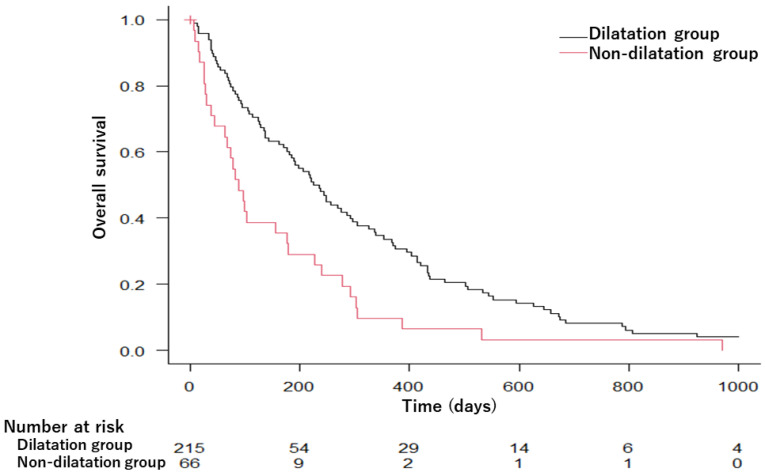
Kaplan–Meier survival curves comparing overall survival in the dilatation and non-dilatation groups. MST was 230 days in the dilatation group and 88 days in the non-dilatation group (*p* = 0.001). MST, median survival time.

**Figure 3 diagnostics-13-00963-f003:**
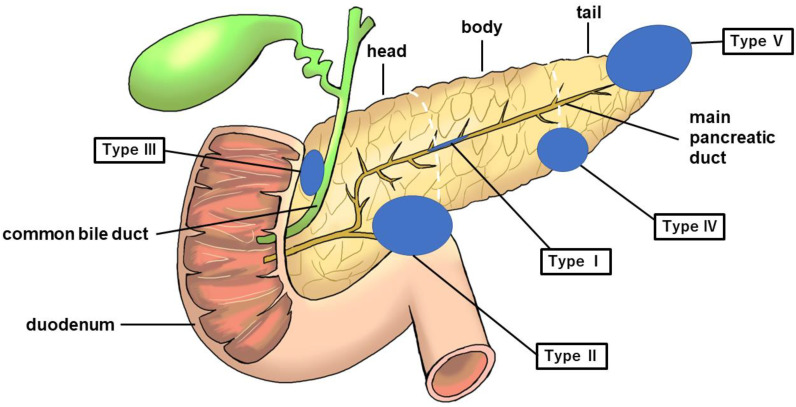
Figure showing five categories of PDAC without MPD dilatation. The blue circle represents the tumor. PDAC, pancreatic ductal adenocarcinoma; MPD, main pancreatic duct.

**Figure 4 diagnostics-13-00963-f004:**
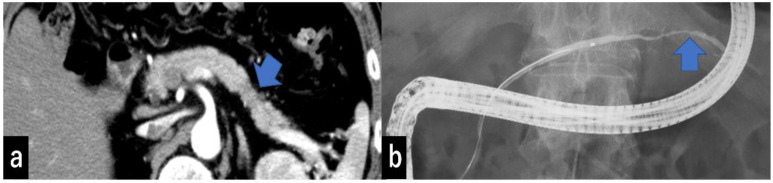
(**a**) CECT shows focal pancreatic atrophy in the pancreatic tail (arrow). (**b**) ERCP shows narrowing of the MPD at the pancreatic tail (arrow) but no MPD dilatation of more than 3 mm. CECT, contrast-enhanced computed tomography; ERCP, endoscopic retrograde cholangiopancreatography; MPD, main pancreatic duct.

**Figure 5 diagnostics-13-00963-f005:**
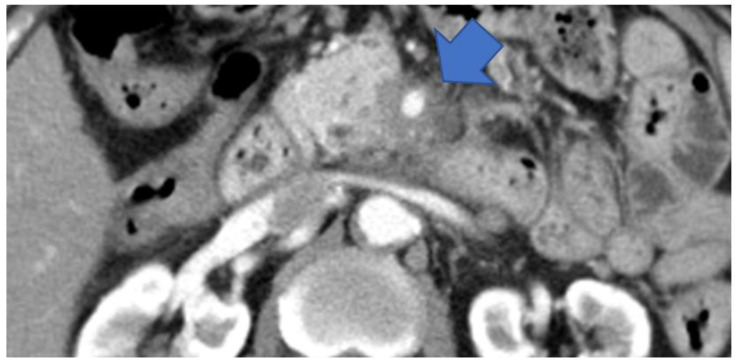
CECT shows tumors surrounding the SMA in the pancreatic uncus area (arrow). CECT, contrast-enhanced computed tomography; SMA, superior mesenteric artery.

**Figure 6 diagnostics-13-00963-f006:**
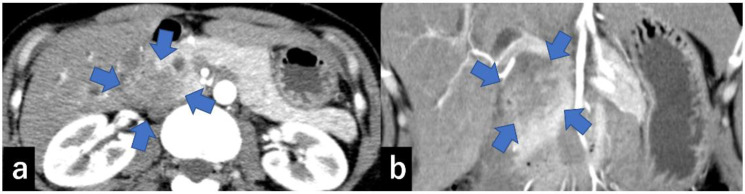
CECT shows a 30 mm large low-absorption area (arrow) in the pancreatic groove area ((**a**) axial, (**b**) coronal). CECT, contrast-enhanced computed tomography.

**Figure 7 diagnostics-13-00963-f007:**
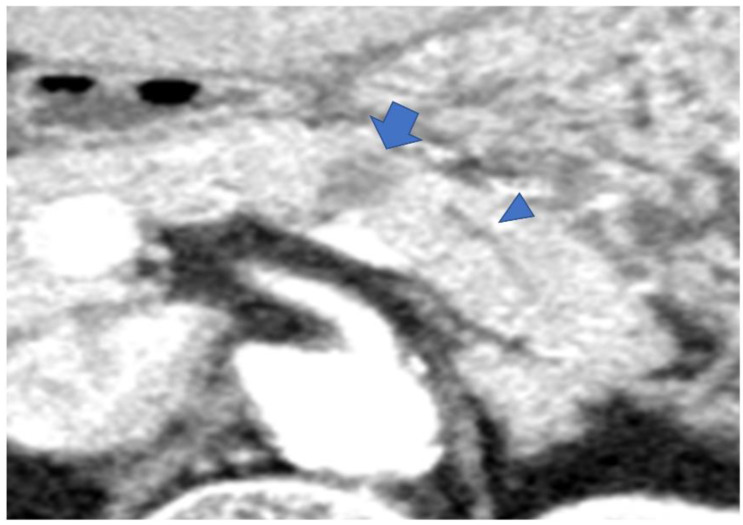
CECT shows an 18 mm large low-absorption area (arrow) in the pancreatic body area. MPD is not dilatated (arrowhead). MPD, main pancreatic duct.

**Figure 8 diagnostics-13-00963-f008:**
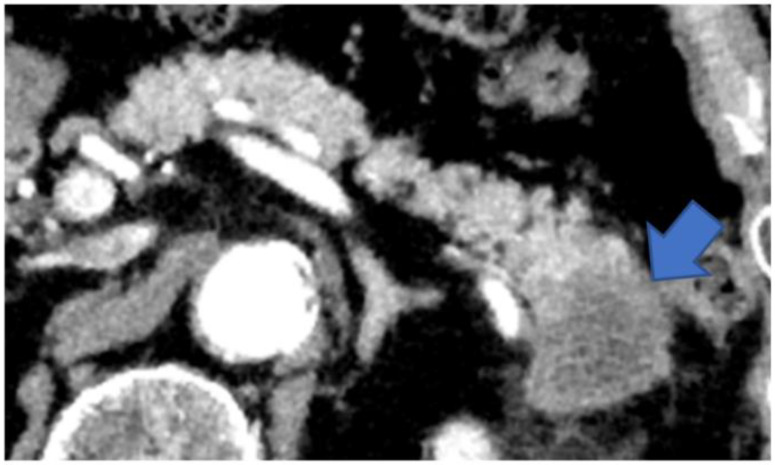
CECT shows a tumor in the pancreatic tail (arrow). The tumor has replaced the pancreatic tail, leaving no space for the MPD to expand. CECT, contrast-enhanced computed tomography; MPD, main pancreatic duct.

**Figure 9 diagnostics-13-00963-f009:**
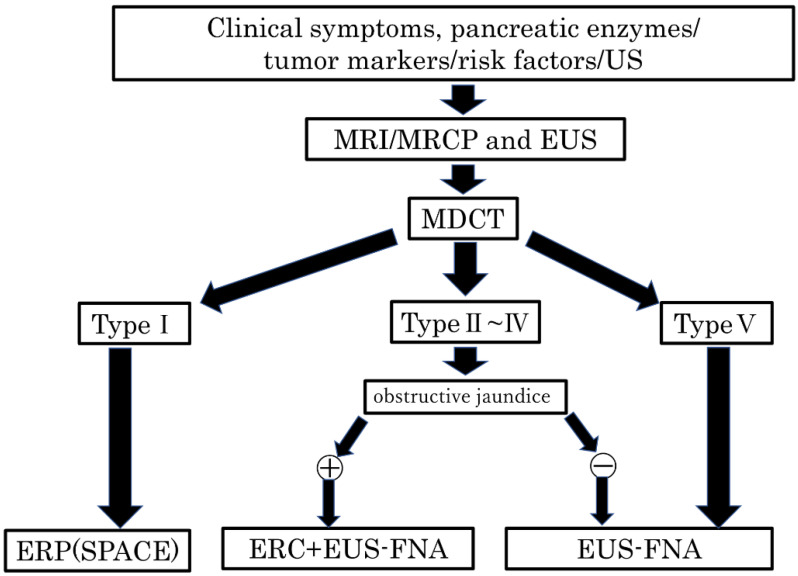
Algorithm of the diagnostic management of PDAC without the MPD dilatation. US, ultrasonography; MRI, magnetic resonance imaging; MRCP, magnetic resonance cholangiopancreatography; EUS, endoscopic ultrasonography; MDCT, multidetector raw CT; ERP, endoscopic retrograde pancreatography; SPACE, serial pancreatic juice aspiration cytologic examination; ERC, endoscopic retrograde cholangiography; EUS-FNA, endoscopic ultrasonography-guided fine needle aspiration; MPD, main pancreatic duct; PDAC, pancreatic ductal adenocarcinoma.

**Table 1 diagnostics-13-00963-t001:** Clinical characteristics.

	Dilatation Group (*n* = 215)	Non-Dilatation Group (*n* = 66)	*p*-Value
Sex, *n* (male/female)	124/91	39/27	0.82
Age, mean ± SD (range)	74 ± 7 (43–97)	74 ± 5 (45–94)	0.9
Tumor size, mm (range)	30.1 (17–43)	33.7 (18–48)	0.75
**Location**			
head/body/tail, *n* (%)	141 (66)/59 (27)/15 (7.0)	22 (33)/4 (6)/40 (61)	<0.001
**Risk factors, *n* (%)**			
DM	50 (23)	19 (29)	0.62
Tobacco use	53 (25)	17(26)	0.87
IPMN	36 (17)	13 (20)	0.27
Chronic pancreatitis	9 (4.2)	1 (1.5)	0.13
Heavy alcohol consumption	41 (19)	12 (18)	0.85
Obesity (>BMI 30 kg/m^2^)	5 (2.3)	2 (3.0)	0.92
Family history of pancreatic cancer	7 (3.2)	2 (3.0)	1

Data are expressed as the number (percentage) or the mean ± standard deviation. * Some patients had multiple risk factors. SD, standard deviation; DM, diabetes mellitus; IPMN, intraductal papillary mucinous neoplasm; BMI, body mass index.

**Table 2 diagnostics-13-00963-t002:** Opportunities for medical examination.

Examination Opportunities	Dilatation Group (*n* = 215) (%)	Non-Dilatation Group (*n* = 66) (%)	*p*-Value
**Symptoms**	130/215 (60)	40/66 (61)	0.77
Abdominal pain	54/130 (42)	16/40 (40)	1
Back pain	10/130 (7.7)	2/40 (5.0)	1
Nausea	9/130 (6.9)	2/40 (5.0)	1
Diarrhea	3/130 (2.3)	2/40 (5.0)	1
Jaundice	52/130 (40)	6/40 (15)	0.014
Weight loss	1/130 (0.77)	4/40 (10)	0.008
Other	24/130 (18)	10/40 (25)	0.052
**Abnormalities identified on medical check-up**	14/215 (6.5)	8/66 (12)	1
Abnormal findings on US	12/14 (86)	3/8 (38)	0.055
Elevated tumor marker levels	1/14 (7.1)	4/8 (50)	0.11
Others	1/14 (7.1)	1/5 (20)	0.4
**Abnormalities identified during screening for other diseases**	58/215 (27)	15/66 (23)	0.74
Abnormal imaging findings	54/58 (92)	15/15 (100)	1
Plain CT	9/54 (17)	7/15 (47)	0.088
CECT	26/54 (48)	7/15 (47)	0.47
US	18/54 (33)	0/15 (0)	0.25
MRI	0/54 (0)	0/15 (0)	1
EUS	1/54 (1.9)	0/15 (0)	0.055
PET–CT	0/54 (0)	1/15 (6.7)	0.21
Elevated pancreatic enzymes	2/58 (3.4)	0/15 (0)	1
Elevated tumor marker levels	2/58 (3.4)	1/15 (6.7)	0.46
**Abnormalities during follow-up of pancreatic diseases**	10/215 (4.7)	3/66 (4.5)	0.75
**Other**	1/215 (0.47)	0/66 (0)	1

Data are expressed as the number (percentage). CT, computed tomography; CECT, contrast-enhanced computed tomography; US, ultrasonography; MRI, magnetic resonance imaging; EUS, endoscopic ultrasonography; PET–CT, positron emission tomography–computed tomography.

**Table 3 diagnostics-13-00963-t003:** Imaging findings.

	Dilatation Group (*n* = 215) (%)	Non-Dilatation Group (*n* = 66) (%)	*p*-Value
Imaging performed, US/CT/MRI/EUS	151/207/133/179	48/54/37/49	
**Detection of pancreatic tumors**			
US	129/151 (85)	20/48 (42)	<0.001
CECT	195/207 (94)	50/54 (93)	0.75
MRI T1	93/133 (70)	24/37 (65)	0.85
MRI T2	78/133 (59)	18/37 (49)	0.58
DW-MRI	119/133 (89)	32/37 (86)	0.8
EUS	178/179 (99)	48/49 (98)	0.38
**Indirect imaging findings**			
**MPD dilatation**			
US	106/151 (70)		
CECT	197/207 (95)		
MRI (MRCP)	122/133 (92)		
EUS	161/179 (90)		
**MPD stenosis**			
US	89/151 (59)	0/48 (0)	<0.001
CECT	194/207 (94)	0/54 (0)	<0.001
MRI (MRCP)	118/133 (89)	1/37 (2.7)	<0.001
EUS	156/179 (87)	2/49 (4.1)	<0.001

Data are expressed as the number (percentage). US, ultrasonography; CECT, contrast-enhanced computed tomography; MRI, magnetic resonance imaging; DW-MRI, diffusion-weighted magnetic resonance imaging; EUS, endoscopic ultrasonography; MPD, main pancreatic duct; MRCP, magnetic resonance cholangiopancreatography.

**Table 4 diagnostics-13-00963-t004:** Histopathological diagnosis.

		Dilatation Group (*n* = 215) (%)	Non-Dilatation Group (*n* = 66) (%)	*p*-Value
ERCP		131/215 (61)	14/66 (21)	<0.001
	Biopsy of CBD stenosis	30/61 (49)	4/6 (67)	0.67
	Brushing cytology of CBD stenosis	30/79 (38)	3/11 (27)	0.74
	ENBD	20/73 (27)	0/6 (0)	0.33
	Single aspiration of pancreatic juice	15/37 (41)	0/1 (0)	1
	Brushing cytology of MPD stenosis	27/47 (57)	1/1 (100)	1
	SPACE	18/32 (56)	0/2 (0)	0.21
Confirmation of malignancy		82/131 (63)	8/14 (57)	0.78
EUS-FNA		154/215 (72)	45/66 (68)	0.36
	Biopsy	142/154 (92)	44/45 (98)	0.3
Gastrointestinal biopsy		20/21 (95)	7/7 (100)	1
Liver tumor biopsy		3/4 (75)	4/4 (100)	1
Cytology of ascites		3/6 (50)	5/7 (71)	0.59

Data are expressed as the number (percentage). ERCP, endoscopic retrograde cholangiopancreatography; CBD, common bile duct; ENBD, endoscopic nasoboliary drainage; MPD, main pancreatic duct; SPACE, serial pancreatic juice cytologic examination; EUS-FNA, endoscopic ultrasonography guided fine-needle aspiration.

**Table 5 diagnostics-13-00963-t005:** (**A**) Clinical stage. (**B**) Resectability classification.

(A)			
Clinical Stage	Dilatation Group (*n* = 215) (%)	Non-Dilatation Group (*n* = 66) (%)	*p*-Value
0	3 (1.4)	1 (1.5)	1
IA	5 (2.3)	2 (3.0)	0.67
IB	8 (3.7)	1 (1.5)	0.69
IIA	67 (31)	10 (15)	<0.001
IIB	12 (5.6)	2 (3.0)	0.74
III	41 (19)	8 (12)	0.33
IV	79 (37)	42 (64)	<0.001
(**B**)			
**Resectability** **Classification**	**Dilatation Group** **(*n* = 215) (%)**	**Non-Dilatation Group** **(*n* = 66) (%)**	***p*-Value**
R	62 (29)	14 (21)	0.16
BR-PV	34 (16)	1 (1.5)	0.004
BR-A	5 (2.3)	2 (3.0)	1
UR-LA	35 (16)	7 (11)	0.68
UR-M	79 (37)	42 (64)	<0.001

Data are expressed as the number (percentage). Clinical stage based on the National Comprehensive Cancer Network Clinical Practice Guidelines in Oncology (version 1, 2020) (12). Data are expressed as the number (percentage). Resectability classification based on the 8th edition of the Union for International Cancer Control (13). R, resectable; BR-PV, borderline resectable with portal vein invasion; BR-A, borderline resectable with arterial invasion; UR-LA, unresectable locally advanced; UR-M, unresectable with metastasis.

**Table 6 diagnostics-13-00963-t006:** Univariate and multivariate analyses of the factors associated with OS.

	Univariate Analysis		Multivariate Analysis	
	Hr (95% Ci)	*p*-Value	Hr (95% Ci)	*p*-Value
**Tumor location**				
Head/Body	1		1	
Tail	1.59 (1.11–2.25)	0.011	1.28 (0.78–2.09)	0.331
**Jaundice**				
No	1			
Yes	1.15 (0.75–1.76)	0.515		
**Weight loss**				
No	1			
Yes	0.38 (0.05–2.57)	0.306		
**Clinical stage**				
0-I	1		1	
II–III	1.60 (0.69–3.71)	0.268	1.28 (0.51–3.18)	0.599
IV	2.05 (1.42–3.00)	0.016	2.83 (1.00–8.00)	0.049
**Resectability classification**				
R	1		1	
BR-PV or BR-A	1.54 (0.89–2.66)	0.124	1.58 (0.86–2.89)	0.138
UR-LA or UR-M	2.19 (1.48–3.22)	<0.001	1.25 (0.65–2.42)	0.503
**Treatment**				
Surgery	1		1	
Chemotherapy or CRT	1.58 (0.97–2.58)	0.068	0.74 (0.41–1.34)	0.318
Best supportive care	4.20 (2.63–6.70)	<0.001	2.92 (1.73–4.93)	<0.001
**Presence of MPD dilatation (** **≥3 mm)**				
Yes	1		1	
No	0.51 (0.34–0.78)	0.001	0.77 (0.53–1.14)	0.753

OS, overall survival; HR, hazard ratio, CI, confidence interval, R, resectable; BR-PV, borderline resectable with portal vein invasion; BR-A, borderline resectable with arterial invasion; UR-LA, unresectable locally advanced; UR-M, unresectable with metastasis; CRT, chemoradiotherapy; MPD, main pancreatic duct.

**Table 7 diagnostics-13-00963-t007:** The classification of 66 cases of PDAC in the non-dilatation group using our classification method.

Type	I	II	III	IV	V
*n*	1	10	11	4	40
Bile duct obstruction	0	4	8	1	0
Duodenal invasion	0	4	4	0	0
Duodenal obstruction	0	4	1	0	0

## Data Availability

The datasets used in the current study are available from the corresponding author upon reasonable request.

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
