# Peer review of "Clinical Features and Prognostic Impact of Pancreatic Ductal Adenocarcinoma without Dilatation of the Main Pancreatic Duct: A Single-Center Retrospective Analysis"

_diagnostics, 2023, doi:10.3390/diagnostics13050963_

Round 1

Reviewer 1 Report (Previous Reviewer 1)

In the comparison to the previous version of the manuscript, the value of the article was improved.

Author Response

Responses to the Comments by the reviewer 1:

In the comparison to the previous version of the manuscript, the value of the article was improved.

Reply:

Thank you very much for your invaluable comments. Thanks to your excellent comments or suggestions, we were able to brush up on the manuscript.

Reviewer 2 Report (New Reviewer)

The authors analyzed pathologically diagnosed PDAC cases divided into two groups, with and without MPD dilatation, and compared their clinical findings, including prognosis. In addition, they extracted factors related to the prognosis of PDAC. Albeit, I consider these findings to provide new insight into cancer-related fields, I still have some suggestions.

1, Most figures and tables are highly professional; however, the authors should guide the readers to the meaning of the images and tables appropriately; otherwise, it is likely to cause misunderstandings. Therefore, I suggest the author consider revising these figures and table legends again.

2, The author demonstrated that constructing a diagnostic system centered on EUS and MRI DWI is necessary for early diagnosis of PDAC without MPD dilatation, which can improve PDAC prognosis. However, It would be much better if the authors could provide Graphical Abstract for this research; I suggest that they can take a look at the recent paper in MDPI (PMID: 31331013, 34834441, 35625729)

3, Line147:… Percutaneous needle biopsy was performed under echo guidance when a metastatic liver tumor was suspected….Since most references are out of date, and the author only cited three papers that were published in 2021, not 2022 or 2023, therefore I suggest the author can discuss more PDAC or metastasis-related research in the current manuscript (PMID: 36673023, 34829830, 34439375)

4, There are a lot of grammatical errors and inappropriate words in the manuscript. It is recommended to send it to a professional language editing company for polishing. Otherwise, it will be difficult for readers to understand the principle clearly.

5, There are few typo issues for the authors to pay attention to; please also unify the writing of scientific terms. “Italic, capital”? For example, Line 78: ….following outocomes.:…..please check “outocomes” or “outcome”?

Author Response

This manuscript is a resubmission of an earlier submission. The following is a list of the peer review reports and author responses from that submission.

Round 1

Reviewer 1 Report

The article revealed the features of PDAC without the dilatation of MPD. Interestingly, it was performed comparising with features of the PDAC concomitant with the dilatation of MPD. It is very important to detect PDAC early. Moreover, authors proposed new classification of PDAC without the dilatation of MPD facilitating its diagnosis and further management. The information and evidences contained in the article are very valuable and may be useful for clinicians, including gastroenetrologists. I recommend the acceptance of this article after revision taking into account following points:

1) In the lines 27-29, authors described pancreatic cancer (PC) as the fourth leading cause of cancer-related deaths in 2019 based on the data from Vital Statistics from the Ministry of Health, Labour, and Welfare. Please emphasaze that this data refer to Japan.

2) 59 patients were excluded at the initial stage of study, because the have not patholoigcal evidence. Please explain what is pathological evidence.

3) Your work proposed new classification of PDAC without the MPD dilatation and emphasized the typical features of the disease. Therefore, it may be basis for the development of further recommendation of the PDAC without the MPD dilatation management. That is why, some issues of your work should be expanded. Why is the tail of pancreas preferred location for PDAC without the MPD dilatation? Please include the potential cause/causes in the Discussion section.

4) Why is the weight loss more dominant features in PDAC without the MPD dilatation compared to the PDAC with the MPD dilatation? Please include the potential cause in the Discussion section.

5) Basing on your results, 39% patients in the non-dilatation group have not symptoms. I think that it should be emphasized in additional paragraph of the Discussion section that PDAC without the MPD dilatation may be detected as pancreatic incidentaloma. In consequence, the asymptomatic development of PDAC without the MPD dilatation may lead to late diagnosis and the diagnosis at the clinical stage of IV resulting from insidious development of the disease. In addition, it may responsible for worse condition of the patients with PDAC without the MPD dilatation comapred to the subjects with the MPD dilatation.    

6) Please indicate a source of the Figure 3 in the description of this Figure in the case of the lack of doing this Figure by your research group.

7) Basing on the results and proposed your new classification, please propose and create the scheme of algorithm of the diagnostic management of PDAC without the MPD dilatation.

8) What may be role of fine needle aspiration in the diagnosis of PDAC without the MPD dilatation? Please include information about it in the Discussion section.

Reviewer 2 Report

It is challenging for me to understand the clinical relevance of this paper. Even though the number of patients is large and the results could be considered potentially relevant, the interest in the results of this retrospective study is very questionable. In my opinion, the main msg that the patients with pancreatic cancer in the tail have less frequency PD dilation is something expected. Still, irrelevant as well as that patients with PDAC in TOP have a more advanced tumor at diagnosis, which has been very well-known for years and has been related to the absence of jaundice. 

Round 2

Reviewer 2 Report

My opinion continues to be the same. It is challenging for me to understand the clinical relevance of this paper. Even though the number of patients is large and the results could be considered potentially relevant, the interest in the results of this retrospective study is very questionable. In my opinion, the main msg that the patients with pancreatic cancer in the tail have less frequency PD dilation is something expected. Still, irrelevant as well as that patients with PDAC in TOP have a more advanced tumor at diagnosis, which has been very well-known for years and has been related to the absence of jaundice.